# The Role of Platelet-Activating Factor and Magnesium in Obstetrics and Gynecology: Is There Crosstalk between Pre-Eclampsia, Clinical Hypertension, and HELLP Syndrome?

**DOI:** 10.3390/biomedicines11051343

**Published:** 2023-05-02

**Authors:** Nidhi Chawla, Hinal Shah, Kathleen Huynh, Alex Braun, Hanna Wollocko, Nilank C. Shah

**Affiliations:** Touro College of Osteopathic Medicine, 60 Prospect Ave, Middletown, NY 10940, USA; nchawla@student.touro.edu (N.C.);

**Keywords:** platelet activating factor, magnesium, pre-eclampsia, clinical hypertension, HELLP syndrome

## Abstract

Clinical hypertension is a complex disease of the cardiovascular system that can affect the body’s ability to physiologically maintain homeostasis. Blood pressure is measured as systolic pressure of the heart and diastolic pressure. When the systolic pressure exceeds values of 130–139 and diastolic exceeds 80–89, the body is in stage 1 hypertension. A pregnant woman with hypertension is predisposed to developing pre-eclampsia during gestation between the 1st and 2nd trimester. If the symptoms and changes in the mother’s body are not controlled, this can progress to hemolysis, elevated liver enzymes, and low platelet count also known as HELLP syndrome. The onset of HELLP syndrome generally begins before the 37th week of pregnancy. Magnesium is one of the most used cations in clinical medicine with various implications in the body. With a critical role in vascular smooth muscle, endothelium, and myocardial excitability it is used in treatment of clinical hypertension, pre-eclampsia in gestational periods, and HELLP syndrome. Platelet-activating factor (PAF) is an endogenous phospholipid proinflammatory mediator that is released in response to various biological and environmental stressors. When released it causes platelets to aggregate, further exacerbating hypertension. The purpose of this literature review is to investigate the role that magnesium and platelet-activating factors have on clinical hypertension, pre-eclampsia, and HELLP syndrome while focusing on the interplay between these molecules.

## 1. Introduction

Hypertension is a common medical condition with elevated systolic blood pressure that affects a significant number of women and has numerous adverse outcomes on maternal and fetal health. Without the proper precautions and interventional measures, hypertension can further escalate to more dangerous conditions such as higher-grade clinical hypertension, pre-eclampsia, and HELLP (hemolysis, elevated liver enzymes, and low platelet count) syndrome. Pre-eclampsia and HELLP syndrome are among the leading causes of fetal and maternal hospitalizations and mortality. Pre-eclampsia is a hypertensive condition that is characterized by the presence of protein in the urine. If left unresolved it can lead to systemic implications throughout the body. HELLP syndrome is identified as increased liver enzymes and decreased platelet counts resulting in hemolysis. 

The molecular functioning of magnesium as a powerful mediator in the progression of obstetric and gynecologic conditions is explored in this review. In an attempt to maintain homeostatic physiology, magnesium can be used as treatment in hypertension, pre-eclampsia, and HELLP syndrome. Platelet-activating factor is a phospholipid mediator that increases in concentration during inflammatory events such as gestational clinical hypertension and exacerbates the condition by encouraging platelet aggregation. We will discuss the role of PAF in physiology and pharmacology, and then introduce magnesium, and ultimately analyze the role that these two mediators play in hypertension, pre-eclampsia, and HELLP syndrome. 

## 2. Platelet-Activating Factor Normal Physiology and Pharmacological Use

PAF is a proinflammatory phospholipid that affects many different pathological and physiological processes. PAF is produced from a subclass of phosphatidylcholines with an ether bond at the sn-1 position on the glycerol backbone [1]. PAF acts via G protein activating phosphatidylinositol-specific phospholipase C [2,3]. PAF has a role in embryogenesis because it stimulates endothelial cell migration and angiogenesis and has been seen to affect cardiac function because it exhibits mechanical and electrophysiological actions on cardiomyocytes [3]. PAF may also have a role in modulation of blood pressure, mainly by affecting the renal vascular circulation, and has a minor role as a precursor for low-density lipoprotein [1,3]. In pathological conditions, PAF is involved in cardiac dysfunction such as cardiac anaphylaxis, hemorrhage, and septic shock. PAF is involved in the recruitment of leukocytes in inflamed tissue by promoting adhesion to the endothelium and extravascular transmigration of leukocytes and causing hypotension under normal levels [2,3]. The activation of PAF in plasma is inhibited by the ability of serum proteins to bind PAF and the presence of a specific hydrolase, platelet-activating factor acetyl hydrolase, and causes platelet activation, which in turn causes the release of thromboxane A2 [4,5]. 

## 3. Magnesium Physiology and Pharmacological Use

Magnesium is primarily found in bone, muscle, and neural tissue, and is the fourth most-prevalent intracellular cation in the body. Less than 1% of the magnesium found in the body is seen in plasma and red blood cells [6,7]. It is seen in the bound form, mostly bound to albumin. Magnesium is a vital cation for many processes in the body, activating more than 300 enzyme reactions such as carbohydrate metabolism, protein, ATP, DNA, RNA synthesis, and oxidative phosphorylation [6,8,9]. Magnesium also regulates muscle contraction such as vascular smooth muscle contraction, neuromuscular conduction, glycemic control, blood pressure, and bone development [7].

## 4. Mg Metabolism

Normal total body Mg content in an adult is 25 g or 2000 mEQ. Magnesium homeostasis is regulated by the intestines, bones, and kidneys [7]. Up to 76% of magnesium ingested is absorbed; the majority of magnesium is absorbed passively in the ileum and distal jejunum, some being absorbed actively in the large intestine, with the rest excreted in the feces. The kidney is the main regulator for magnesium. The majority of excreted magnesium is reabsorbed in the thick ascending limb of the loop of Henle and distal tubules, which is dependent on the plasma magnesium concentration [6,7,9]. Excess plasma calcium activates calcium receptors in the kidney to increase elimination of the ions. Therefore, induced magnesium induces its own clearance, which is an important safety factor if there is normal renal function [6,10]. A total of 60% of magnesium is in bone, with 30% of that functioning as a reservoir to stabilize the serum concentration, 20% is in skeletal muscle, 19% is in soft tissue, and the remaining amount is in the extracellular fluid as shown in Figure 1. Less than 1% of magnesium is in the serum, and in normal adults the total serum amount of magnesium ranges between 0.70 and 1.1 mmol/L as shown in Table 1. A total of 20% of serum magnesium is protein-bound; of that 20%, 70% is associated with albumin and the rest is bound to globulins, and 15% is complexed with anions such as phosphate and citrate [10]. The remaining 65% is ionized.

## 5. Mg-Deficiency General

The consequences of magnesium deficiency include chronic fatigue states, delirium, muscle weakness, glucose metabolism disorder, myocardial arrhythmias, and electrolyte disturbances [6]. As stated above, the normal range of total serum Mg2þ is 0.75–1.0 mmol; however, as magnesium is an intracellular ion, whole body magnesium depletion could exist even when there are normal or elevated plasma Mg2þ concentrations [6,10]. Deficiency is correlated with an increase in interleukin-1, tumor necrosis factor-α, interferon-γ, and substance P, which are inflammatory responses in the body [9]. Magnesium deficiency in rats had been correlated with greater inflammatory response.

## 6. Mg and Its Role in Activating PAF

It has been suggested that Mg deficiency leads to an increase in PAF and PAF-like molecules [1,11]. PAF may be an indicator for the etiology of inflammatory conditions. Low-Mg environments can be a trigger that induces hypoxia and ischemia, which induces inflammatory responses such as the release of PAF [1,12]. Mg is involved in over 300 different enzymatic reactions and it has been suggested that Mg has an effect on nuclear factor-kappa B (NF-κB), a regulator of growth, differentiation, cell migration, and cell death as shown in Figure 2. It is also involved with inflammatory response and hypertension [13]. It has been seen that in environments with short-term Mg deficiency, there is an increase in NF-κB in cardiovascular tissues and vascular smooth muscle [1]. As previously stated, an increase in PAF and a decrease in Mg has been seen in pregnancies with hypertension and pre-eclampsia, and those with HELLP syndrome. It has been suggested that the link between Mg deficiency and the increasing circulating levels of PAF is the major trigger in development of these conditions. 

## 7. The Role of PAF in Hypertension

Decreased PAF levels as a result of increased platelet-activating factor acetylhydrolase activity was seen in pregnant patients with hypertension. It has been seen that in pregnant patients with normal blood pressure, the activity of the enzyme decreased to 35% [14]. In comparison to pregnant women with normal blood pressure, the activity of the enzyme in pregnant patients with hypertension was higher. Males had the highest PAF-acetyl hydrolase activity at 65 nmol × min^−1^ × mL^−1^ plasma, and normal pregnancies had the lowest PAF-acetyl hydrolase activity at 30 nmol × min^−1^ × mL^−1^ plasma. Non-pregnant women and pregnancy-induced hypertension had similar levels near 40 nmol × min^−1^ × mL^−1^ plasma [14]. Higher PAF acetylhydrolase activity results in a lower plasma PAF level, and ultimately a lower PAF level could result in vasoconstriction and hypertension, as seen in Figure 3.

## 8. The Role of Mg Deficiency in Hypertension

Evidence suggests a linkage between magnesium deficiency and hypertension described as an inverse relationship between intracellular magnesium concentration and blood pressure [15,16]. As stated before, magnesium regulates vascular muscle contraction and indirectly influences blood pressure regulation [15,17]. As shown in Figure 4, with decreased magnesium there is an increase in intracellular calcium, which results in increased vasoconstriction and hypertension. It has been seen that administering magnesium induces hypotension and vasodilation of blood vessels. In vascular smooth muscle cells, magnesium acts extracellularly by inhibiting the transmembrane calcium transport, which inhibits calcium entry causing a decrease in contractile actions; alternately, magnesium acts intracellularly as a calcium antagonist, which in turn modulates the vasoconstrictor action of the increased amount of calcium. There is a clear connection between magnesium levels and calcium in regards to vascular contractions, tone, and hypertension.

## 9. Pre-Eclampsia

Pre-eclampsia is a complication that could occur in late pregnancy, labor, or early parts of delivery, and affects about 3% of women [18]. It is characterized by hypertension, abnormal protein levels in urine, and other systemic disturbances. If not treated, it could progress to eclampsia, which is characterized by seizures and can be fatal to the mother and fetus. There is no way to prevent pre-eclampsia, and the best way to manage it is to screen and induce delivery when needed. Trophoblastic cells from the placenta ultimately contribute to pre-eclampsia. There are two categories of pre-eclampsia: maternal and placental. Placental pre-eclampsia is caused by the placenta being under hypoxic conditions with oxidative stress. Maternal pre-eclampsia is caused by the interaction between a normal placental and a maternal constitution that is susceptible to microvascular disease such as long-term hypertension or diabetes. Mixed presentations are common as well. 

Pre-eclampsia is difficult to diagnose since it can present in numerous different ways and there is not a consistent diagnostic test [19]. As stated before, the fundamental features of pre-eclampsia are new-onset hypertension and proteinuria. However, 20% of women have atypical presentation of pre-eclampsia, making it difficult to diagnose. Women can also develop severe headache, visual changes, acute liver injury, hemolysis, thrombocytopenia, and seizures. Fetal complications from pre-eclampsia include iatrogenic prematurity, fetal growth restriction, and oligohydramnios, and there is an increased risk of prenatal death. Fetal complications are generally believed to be caused by impaired uteroplacental blood flow, placental abruption, and infarctions.

## 10. Pre-Eclampsia and PAF

Pre-eclampsia, a complication that can be life-threatening, can affect multiple organs. It is associated with an increase in platelet function [20]. When platelets become activated the mean platelet volume (MVP) increases. Platelet activation induces changes on the surface of the platelets, causing high levels of aggregation leading to pre-eclampsia. It has been seen that reduced serum inhibitory potential of PAF increases incidence of pre-eclampsia [21]. PAF is a mediator of inflammation causing an increase in thromboxane production, platelet aggregation, and platelet destruction. An elevated level of PAF has been thought to induce hypertension caused by vasoconstriction from increased production of thromboxane [22]. 

In a study conducted by Rowland, Vermillion, and Roudebush to see the link between PAF and pre-eclampsia, it was seen that there were higher PAF concentrations in those with pre-eclampsia when compared to the pregnant group without pre-eclampsia and the non-pregnant group [22]. The PAF concentrations seen in the pregnancies with pre-eclampsia were higher by 35% when compared to pregnancies without pre-eclampsia. The PAF concentrations in the pregnancies with pre-eclampsia were also 29% higher compared to non-pregnant participants of the study. The findings from Rowland, Vermillion, and Roudebush are consistent with those of Benedetto et al. The study conducted by Bendedetto et al. also found that serum inhibitory activity of PAF was significantly lower in patients with pre-eclampsia [21]. This finding correlates with the increase in PAF. It has also been seen that there was an increase in PAF-mediated platelet aggregation in pregnancies with pre-eclampsia [22]. It has been found that in certain cases patients taking a PAF inhibitor such as theophylline have lower incidence of pre-eclampsia [23,24]. There was a reduction in the frequency of incidence and the severity of pre-eclampsia with use of anti-platelet treatments. As shown in Figure 5, prostacyclin has a role in inhibiting platelet aggregation and the levels in pregnancy are lower than those seen in non-pregnant individuals [25]. Analog prostacyclin is given as an anti-platelet treatment to vasodilate the blood vessels by increasing levels of cAMP to lower the frequency of pre-eclampsia [25,26]. 

## 11. The Effect of Mg Deficiency with Pre-Eclampsia

Pre-eclampsia early onset can cause many fetal complications. Magnesium deficiency is seen in women that develop pre-eclampsia and can contribute to impairment of neonatal development and metabolic issues that can extend into adulthood [6,27]. Lower magnesium levels have been seen in red blood cells’ membranes and the brain of pregnant women with pre-eclampsia when compared to women with normal pregnancies [28]. In Table 2 it presented that magnesium sulfate is an inorganic salt that is found in both maternal blood at 0.75 mmol/L and fetal blood at 0.83 mmol/L [27,28]. However, in pregnant women with pre-eclampsia, it decreases to 0.66 mmol/L and in the fetus it increases to 1.01 mmol/L. The increase in magnesium accumulation in fetal circulation in pregnancies with pre-eclampsia indicate there might be altered magnesium metabolism in the fetus. Given the link between magnesium deficiency and pre-eclampsia, intravenous magnesium therapy is used to counteract pre-eclampsia. Intravenous magnesium is thought to work on the endothelium lining of the vascular system through a dilatory mechanism. This reduction in the ischemia produced in vessels helps to counteract the pre-eclampsia that is seen [29]. 

As stated previously, magnesium is a major cofactor in the body that regulates many of the body processes. In pre-eclampsia there is a decrease in magnesium levels, as shown in the study conducted by Sukonpan and Phupong [30]. The study conducted by Sukonpan and Phupong found that compared to pregnancies without pre-eclampsia, pregnancies with pre-eclampsia are presented with a decrease in calcium and magnesium levels. These results are in concordance with other studies, such as that conducted by Purohit et al., which also found that hypo-magnesium is seen in pre-eclampsia [31]. Hypo-magnesium in pregnancy is associated with hemodilution, decreased renal clearance during pregnancy, and decreased consumption of minerals by the fetus [30,31]. It has been seen that magnesium improves the endothelial function in pre-eclampsia since magnesium has a role in vasodilation, inhibiting platelet adherence and aggregation. Therapeutic magnesium sulfate has been the best treatment for hypo-magnesium seen in pre-eclampsia to lower blood pressure. It has been seen that patients with pre-eclampsia taking magnesium sulfate increase the synthesis of prostacyclin to counteract some of the effects of pre-eclampsia by inhibiting platelet aggregation [32]. With the addition of magnesium sulfate to treatment plans for preeclamptic patients, the levels of prostaglandin I2 were found to be enhanced [33]. 

## 12. Clinical Hypertension (HTN) Prior to Pregnancy

Clinical hypertension is a condition in which the blood pressure of an individual is at minimum 140 mmHg systolic pressure or 90 mmHg diastolic pressure at the beginning of pregnancy or during the gestational period [34,35]. Given that there can be discrepancies in blood pressure readings, it is advised that readings be screened at an interval of 4 h to ensure that, in the case of severe hypertension, the diagnosis is not confirmed in an interval deemed too short [36]. There are various stages in the hypertension diagnosis. According to the American College of Cardiology, the most recent recommendation, as of 2017, is that normal blood pressure be less than 120 mmHg systolic pressure and less than 80 mmHg diastolic pressure. The diagnosis of hypertension begins at stage 1 with blood pressure values of systolic pressure between 130 to 139 mmHg and diastolic pressures between 80–89 mmHg. The diagnosis is then escalated to stage 2 hypertension if the values of systolic pressure are greater than 140 mmHg and diastolic pressure exceeds 90 mmHg [34,37]. When levels approach the stage 2 diagnosis, immediate drug intervention is suggested in patients. Depending on the severity of accompanying symptoms with a stage 2 diagnosis, lifestyle modification advice is optimal. However, drug treatment in high-risk patients and those with accompanying diseases such as chronic vascular disease, chronic kidney disease, diabetes mellitus, and hypertension-mediated organ damage can be required [36,37,38]. 

Clinical hypertension is a condition that affects a significant proportion of individuals in the United States given the increasing rates of obesity and metabolic conditions. Clinical hypertension during gestation has been associated with “increased rates of adverse maternal and fetal outcomes” in the acute and long-term scopes [35,38]. Current research shows the rate of occurrence to be around 3%, which has been noted to be increasing over time [34]. From the years of 2000 to 2009 alone there was a dramatic increase in maternal clinical hypertension of nearly 67%, with the greatest demographic of women affected by this condition being African American women [34,36]. With numbers reaching this magnitude, it is becoming of increasing concern because of the obesity epidemic and the shift to later-age pregnancies in the developed world. Numerous maternal risks have been associated with the prior diagnosis of clinical HTN during pregnancy, inclusive of but not limited to stroke, multiple organ failure, vascular coagulation, and placental abruption. The risks are not limited to the mother but can affect the fetus as well, with intrauterine growth retardation, preterm birth, or ultimately intrauterine death [37]. 

## 13. Clinical Hypertension and Mg

Once the diagnosis has been made, hypertensive management techniques must be implemented in order to protect the health of the mother and fetus. Given that hypertension can alter the elastic properties found in the tunica media of arteries, treatments must address the onset of vascular stiffness and hypomagnesemia that are associated with the diagnosis. The hypomagnesemia component can induce endothelial dysfunction, dyslipidemia, and other inflammatory effects within the vascular structures [35,36,39]. Magnesium homeostasis has been extensively studied in relation to hypertension and there have been suggestions that it can prevent gestational hypertension [36]. Magnesium sulfate is considered the first-line therapy in women who are suffering from clinical HTN. Given that experimental models show that there is a reduced concentration of serum magnesium levels in models that have hypertension, the supplementation of magnesium can serve an antihypertensive purpose [39]. 

## 14. Clinical Hypertension and PAF

Platelet-activating factor acetyl hydrolase is a more recently studied endogenous phospholipid that has had increasing relationships with the diagnosis of hypertension and atherosclerosis. This phospholipid has a role in the induction of inflammation, in particular during oxidative stress conditions. PAF is modified under oxidative conditions to produce oxidative stress in utero. In a study conducted by Fan et al., it was seen that neonates have been shown to have a significantly increased amount of plasma PAF-AH when the mother was diagnosed with mild hypertension and an even higher amount of PAF-AH was found in neonates with mothers diagnosed with severe hypertension [40]. Due to the increased amount of inflammation, mothers with severe HT were also found to have higher atherogenic index and triglyceride values, which coincide with the role PAF-AH is found to have physiologically [40]. In a study conducted by Fan et al. it was seen that aside from hypertension, PAF was found to have other adverse effects in pregnancy such as the increased risk of placenta-mediated pregnancy outcomes. These implications are further exacerbated in women who are previously diagnosed with hypertension, and aspirin has been found to partially alleviate the magnitude of the adverse outcomes. Increased maternal levels of plasma platelet factor 4 were found to drive the association between the elevated levels and hypertensive disorder [41]. 

## 15. HELLP Syndrome

HELLP is an acronym used to describe a disease state that ultimately leads to hemolysis, elevated liver enzymes, and low platelets [33,42]. While pre-eclampsia is characterized by the appearance of hypertension parameters and is a major obstetrical issue, HELLP syndrome is an obstetrical emergency and often occurs concurrently with pre-eclampsia during pregnancy [42]. This genetic condition exceeds the pathophysiology of the fetus and can be due to the maternal system’s ability to deal with the pregnancy. Though not common, this condition occurs in 0.5% to 0.9% of all pregnancies and in 10%–20% of cases deemed severe pre-eclampsia [33,42,43]. 

HELLP syndrome is a condition in which high mortality and morbidity rates are present, and there can be life-threatening implications for mother and fetus if the symptoms are not managed [44]. The mortality rate of women with HELLP syndrome is between 0% and 24%, with a perinatal death reaching values as high as 37% [45]. It is suggested to deliver the fetus if the symptoms develop after 34 weeks of gestational age and the health status of mother and fetus are deteriorating even post-intervention [44]. In a study accessing women who were between 23 and 34 weeks of gestational age and diagnosed with HELLP syndrome, expectant management of the condition included induction within 48 h following the diagnosis and resulted in favorable results. Taking the proper expectant measures overall resulted in longer gestation times after diagnosis, with values reaching 7.75 days without increasing maternal or fetal mortality [46]. In order for a complete diagnosis to be made, several laboratory tests need to be established, such as a complete blood count, peripheral smear, and testing of liver function (aspartate aminotransferase and alanine aminotransferase), and creatine levels. If the liver panel tests are found to be elevated further, coagulation studies must be conducted to assess fibrinogen, prothrombin, and activated partial thromboplastin time [45]. 

Although the etiology of the disease is not thoroughly studied, it is thought to be due to a systemic inflammatory disorder that is mediated by a complement cascade. It is postulated that the pathogenesis of HELLP syndrome is like that of pre-eclampsia due to poor placentation leading to increased activation of the complement system. Ultimately, the patient suffers from increased hepatic inflammation and hemolysis and elevated liver enzymes [45,47]. Complement cascade of pre-eclampsia and HELLP with added integration of low Mg2+ levels lead to activation of NF-κB, which in turns activates the complement, as illustrated in Figure 6 [1,45]. 

The diagnosis of this disease is in patients who are usually over the age of 35 years old and the symptoms present between 28 weeks and 37 weeks of gestational age. This time period generally falls within the third trimester. It can also be diagnosed within 7 days postpartum [45]. When urine levels of terminal complement complex C5b-9 are found to be elevated, there is suspicion of deposition in the placental surfaces. When comparing the C5b-9 levels of plasma and urine, women who were diagnosed with pre-eclampsia had significantly higher levels than those experiencing uncomplicated pregnancies. 

## 16. HELLP Syndrome and Magnesium Deficiency

It has been established that a correlation exists between hemolysis and magnesium deficiency [48]. In those with hemolytic anemia, there has been found to be a significant decrease in serum magnesium levels. It has been seen in rats that a low-magnesium diet caused a rapid decrease in hemoglobin levels and a reduced lifespan of red blood cells. As shown Figure 7, magnesium deficiency is also seen in liver diseases, which in turn increases liver enzymes. It has been seen in those with liver cirrhosis that there is a low body magnesium level [49]. Liver diseases influence body magnesium content since albumin is a major transport protein of magnesium. In turn, the imbalanced magnesium levels influence and increase the cirrhosis progression of the liver. A link has been found between thrombocytes and magnesium levels. There is a dose–response relationship between high levels of magnesium, higher counts of platelets, and decreased occurrence of thrombocytopenia or low platelet levels, suggesting serum magnesium is involved in the function of platelets [50].

Once the diagnosis has been made it is recommended for the expectant management to take place by 34 weeks gestation. Complete blood count, comprehensive metabolic panels, and urinalysis and coagulation profiles should be regularly taken. Magnesium sulfate therapy and hypertensive management are highly recommended if the diagnosis is made within the time frame of 24 to 34 weeks [33,51]. It is recommended to administer magnesium sulfate intravenously over 20 min in a loading dose of 6 g until the postpartum period/24 h post-delivery [45]. Magnesium sulfate is the first treatment given to treat HELLP syndrome [45,51]. Magnesium sulfate causes cerebral vasodilation to reduce ischemia in blood vessels, therefore reducing hypertension. 

## 17. HELLP Syndrome and PAF

HELLP syndrome is associated with endothelial injury, deposition of fibrin in the lumen of blood vessels, and increased platelet activation with platelet consumption [5]. As stated before, platelet activation causes the release of thromboxane A2, a vasoconstrictor. A decrease in the amount of circulating platelets and the increased rate of consumption at the sites of damage to vascular endothelium are seen in HELLP syndrome [5,33]. 

## 18. The Crosstalk between Mg and PAF

In pregnancies affected by pre-eclampsia, concentrations of Mg were decreased and those of PAF were increased. The inhibitory factors of PAF were decreased further, increasing PAF concentrations and causing an increase in PAF-mediated platelet aggregation [20,21]. There is a reduction in the incidence of pre-eclampsia with the use of anti-platelet treatments, such as prostacyclin analogs, by causing vasodilation of the blood vessels [24,25,26]. Increased magnesium intake has increased the synthesis of prostacyclin, suggesting that exogenous magnesium should be one of the first courses of action to treat pre-eclampsia [24,25]. As discussed previously, it is to be thought that HELLP syndrome leads to an increase in the complement system, causing an increase in hepatic inflammation, hemolysis, and elevated liver enzymes [45,52]. The complement system is further activated by the low magnesium level activating the NF-κB pathway to further activate the complement system [1,45,47]. Similar to pre-eclampsia, magnesium sulfate is one of the first lines of treatment for HELLP syndrome to cause vasodilation to reduce hypertension.

## 19. Conclusions

Hypertensive conditions are a common complication seen in pregnancies, affecting both maternal and fetal health. When unchecked, they can progress to chronic hypertension, pre-eclampsia, and HELLP syndrome. In this review we discussed the role magnesium as an ion plays in many processes in the body. A decreased concentration in magnesium has many detrimental effects on the body such as hypertension and vasoconstriction. We also discussed the role PAF plays in leading to an increase in inflammation. We noted that magnesium also plays a role as an antihypertensive. Experimental studies have shown that supplementation of magnesium in hypertensive pregnancies resulted in decreased blood pressure. The link between magnesium and the inflammatory response caused by PAF is still of unknown etiology; however, it is generally thought to be linked through the NF-κB path causing high levels of aggregation-inducing hypertension. 

In this review we discussed the detrimental effects of hypomagnesemia and increased levels of PAF during pregnancy. We cohesively brought together the clinical implications of PAF in these obstetric conditions. Through our analysis of the literature, we also connected the two powerful molecules of PAF and Mg, furthering the crosstalk between them. However, the etiology linking the role PAF is not fully understood and further investigation is warranted. Further research should investigate the relationship between inflammatory molecules such as PAF and the resulting hypertensive disorders during pregnancy. Given the current understanding of intravenous Mg utilization in clinical practice, it would be beneficial for research to explore the introduction of dietary Mg in mothers at various gestational ages to determine the point in the timeline that introduction is most crucial. 

## Figures and Tables

**Figure 1 biomedicines-11-01343-f001:**
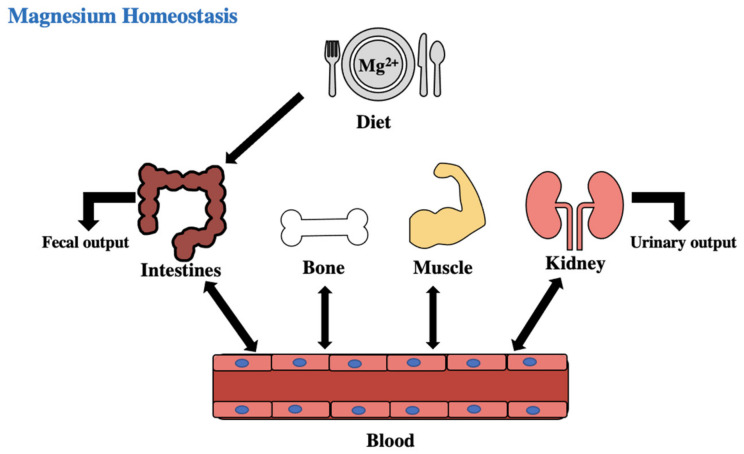
Magnesium balance in the body [7].

**Figure 2 biomedicines-11-01343-f002:**
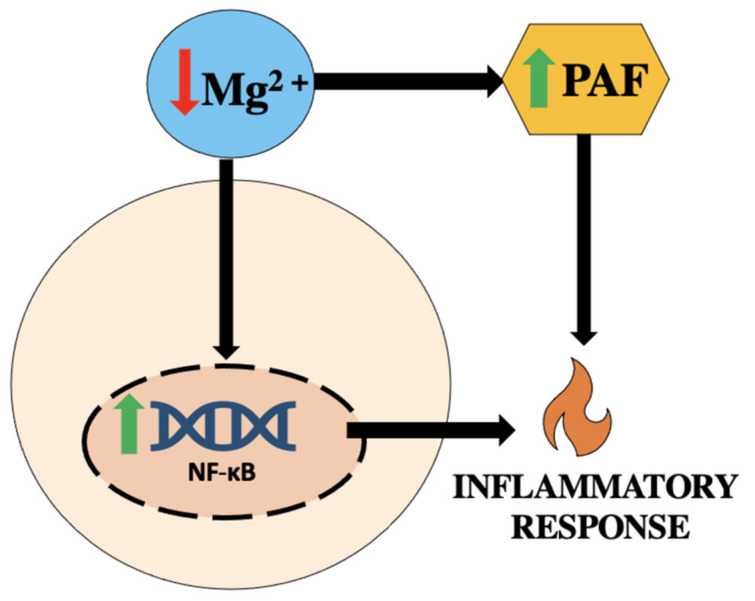
NF-κB role in inflammatory response [1,12].

**Figure 3 biomedicines-11-01343-f003:**
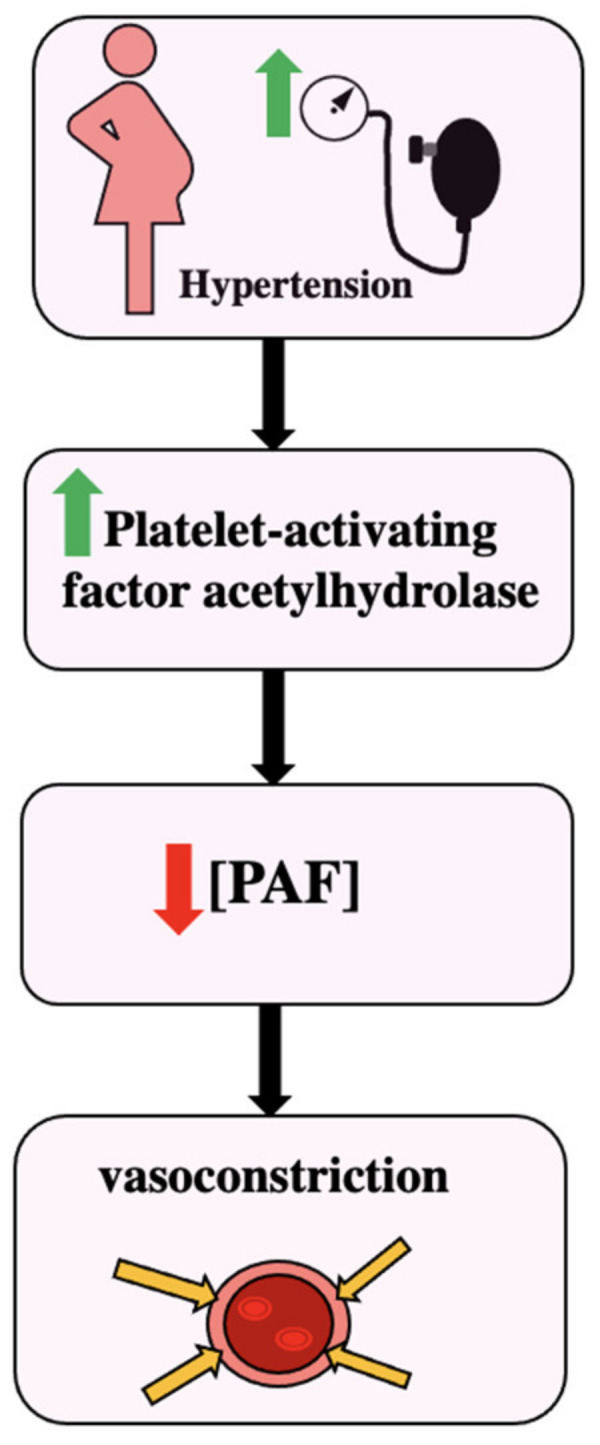
The effect of hypertension and platelet-activating factor acetylhydrolase [14].

**Figure 4 biomedicines-11-01343-f004:**
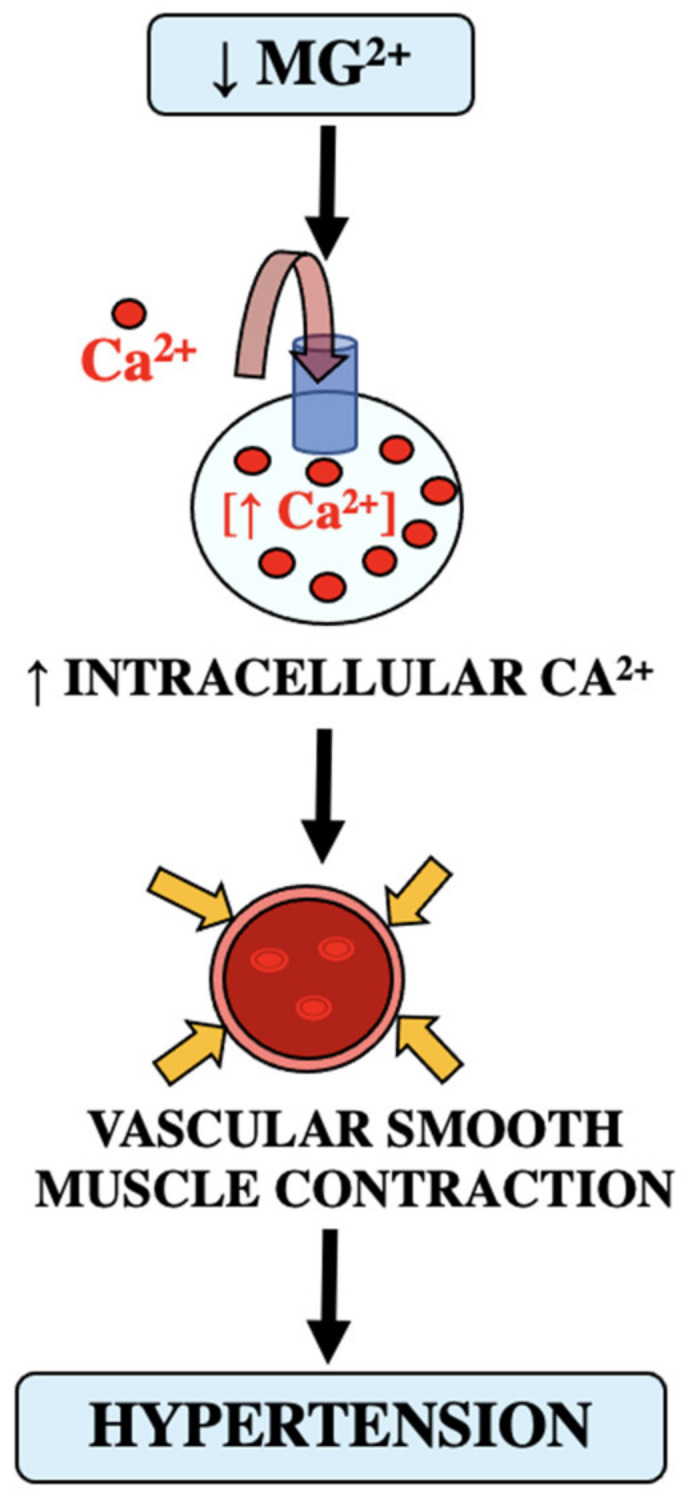
Role of decreased magnesium escalating to increased intracellular calcium increasing vascular smooth muscle contractions [15,16,17].

**Figure 5 biomedicines-11-01343-f005:**
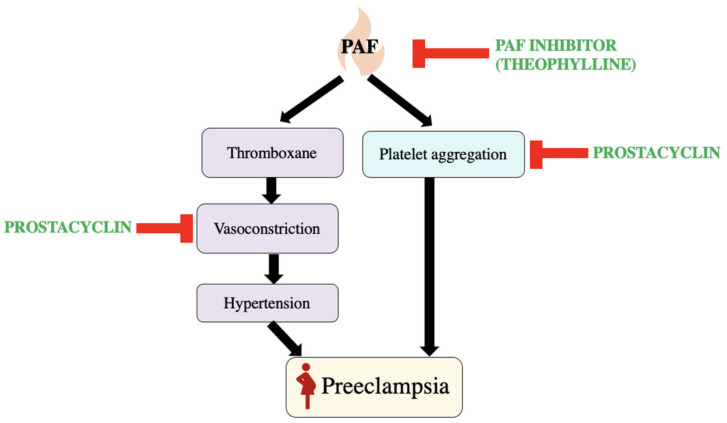
The effect of PAF on the body causing pre-eclampsia and enzymes that inhibit the effects of PAF [25].

**Figure 6 biomedicines-11-01343-f006:**
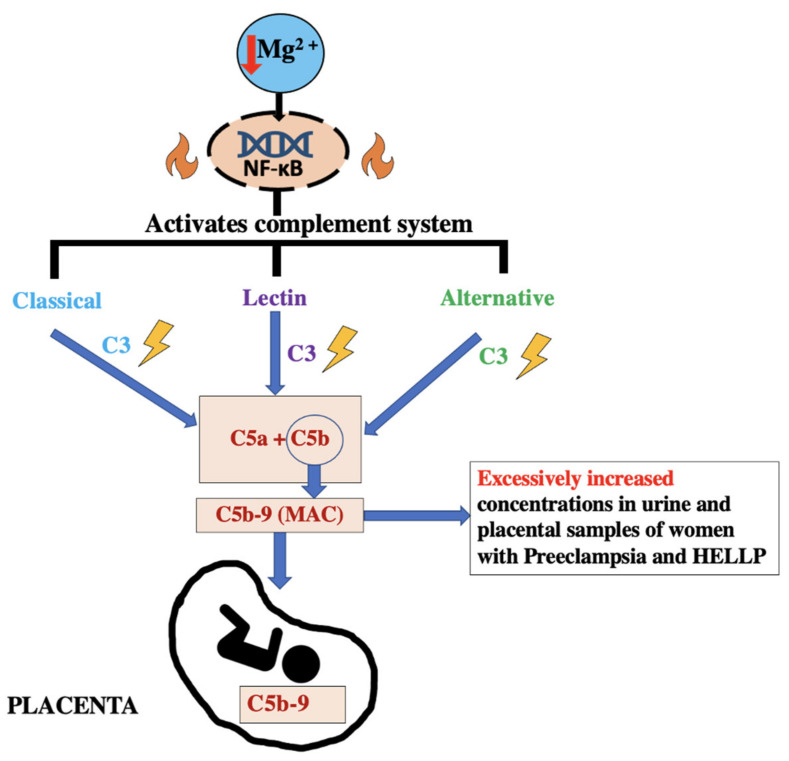
Complement cascade dysregulation in both pre-eclampsia and HELLP syndrome due to decreased Mg2+ [1,42,45].

**Figure 7 biomedicines-11-01343-f007:**
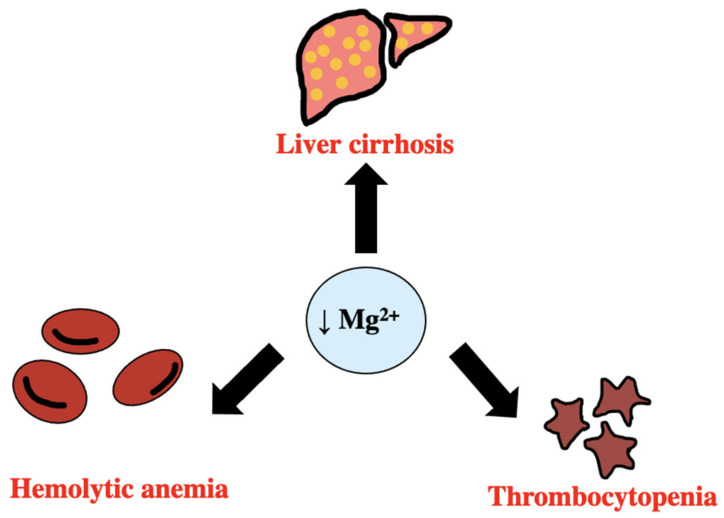
Effects of Mg2+ deficiency leading to liver cirrhosis, hemolytic anemia, and thrombocytopenia [48,49,50].

**Table 1 biomedicines-11-01343-t001:** Ranges of magnesium concentrations in three units [6].

	mg dL^−1^	mEq^−1^	mmol L^−1^
Normal Range	1.8–2.4	1.5–2.0	0.75–1.0
Therapeutic Range	4.8–8.4	4–7	2–3.5
Neuromuscular Toxic Level	>12	>10	>5

**Table 2 biomedicines-11-01343-t002:** Summary of Mg level differences in normal pregnancies vs. pregnancies with pre-eclampsia [27,28].

Magnesium Sulfate Levels (mmol/L)	Normal Pregnancies	Pregnancies with Pre-Eclampsia	Indications
Maternal Blood	0.75 mmol/L	0.66 mmol/L	Decrease in maternal serum magnesium increases risk of pregnancy complications.
Fetal blood	0.83 mmol/L	1.01 mmol/L	Increase in fetal serum magnesium indicates altered metabolism.

## Data Availability

Not applicable.

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
