# Peer review of "The Role of Platelet-Activating Factor and Magnesium in Obstetrics and Gynecology: Is There Crosstalk between Pre-Eclampsia, Clinical Hypertension, and HELLP Syndrome?"

_biomedicines, 2023, doi:10.3390/biomedicines11051343_

Round 1
Reviewer 1 Report
The authors present a manuscript which aims to investigate the role of platelet activating factor in the pathogenesis of hypertensive disorders in pregnancy. The manuscript has been well written but the authors should specify and highlight what this manuscript adds to literature. Otherwise, the manuscript would be a simple reflection of what is already kown about the topic. The authors should discuss about the limitations of their findings and their probable clinical implications in the conclusion part. Moreover, the references that were published before 2008 should be replaced with newr and more up-to-date ones if possible. I think that the manuscript can be accepted for publication after required coorections have been made.
Author Response
Re # MS Revision; biomedicines-2334258
Point-by-Point Response to Reviewer 1 Comments #
Reviewer 1 # The authors present a manuscript which aims to investigate the role of platelet activating factor in the pathogenesis of hypertensive disorders in pregnancy. The manuscript has been well written, but the authors should specify and highlight what this manuscript adds to literature. Otherwise, the manuscript would be a simple reflection of what is already known about the topic. The authors should discuss the limitations of their findings and their probable clinical implications in the conclusion part. Moreover, the references that were published before 2008 should be replaced with newer and more up-to-date ones if possible. I think that the manuscript can be accepted for publication after the required corrections have been made.
Response: Thank you for these insightful comments! We have included in the conclusion areas of research we feel would be beneficial to explore in order to learn more about clinical implications in this area of study. The references that were published before 2008 are foundational to the literature and we found them to be very important in assessing the relationship between PAF and Mg, however in order to make sure the research is up to date we have also included more recent references to our paper.
Reviewer 2 Report
Minor points.
Unify 'PAF' instead of 'platelet activating factor'.
Author Response
Point-by-Point Response to Reviewer 2 Comments #
Reviewer 2 # Minor points.
Unify 'PAF' instead of 'platelet activating factor'.
Response: Thank you for pointing this out. We have consolidated to using PAF after the initial introduction of the molecule.
Reviewer 3 Report
1. The subject is of interest, even if the two molecules (PAF and Mg) are not exactly new kids on the block. However, the organization of the manuscript is utterly chaotic:
- we expect that the two "storylines" (PAF and Mg) will ultimately come together, but they do not, except hypothetically on page 3. The purported "cross talk" between both remains elusive.
- the Title gives a good idea of the chaos: "is there crosstalk between preeclampsia, clinical hypertension and HELLP syndrome?" I would think that blood pressure is the "crosstalk" between the 3 conditions. But the authors probably mean the crosstalk between the 3 clinical conditions and the two molecules. In any case, the formulation is unclear.
- Clinical hypertension (HTN) before pregnancy is usually called "chronic", "pre-existent" or "pregestational" hypertension. We are given an explanation of the definition of hypertension in pregnancy on page 8 (!) instead of the Introduction.
- Pre-eclampsia is defined on page 6. Again, we would expect to find this in the Introduction.
- The headings PAF and pre-eclampsia feature on page 4 and again on page 6, Mg and pre-eclampsia on page 5 and again on page 7.
2. Chronic HTN, pre-eclampsia and the HELLP syndrome are presented like different conditions, which obviously there are not. We would like to know when and how Mg and PAF come into the EVOLVING picture.
3. on Mg: "intravenous Mg therapy is used to counteract preeclampsia and its symptoms" (line 167) and "intravenous magnesium therapy is used to counteract HELLP and its symptoms" (line 185). Mg is used primarily to prevent or treat eclampsia, convulsions, and works at the motor neuron - muscle interface. Strangely for an article on Mg, the authors have made no inquiries on the rationale and the therapeutic history (introduced at Dallas, in fact) of Mg. "Counteract" should suffice.
Author Response
Point-by-Point Response to Reviewer 3 Comments #
Reviewer 3 #
- The subject is of interest, even if the two molecules (PAF and Mg) are not exactly new kids on the block. However, the organization of the manuscript is utterly chaotic:
- we expect that the two "storylines" (PAF and Mg) will ultimately come together, but they do not, except hypothetically on page 3. The purported "cross talk" between both remains elusive.
- the Title gives a good idea of the chaos: "is there crosstalk between pre-eclampsia, clinical hypertension and HELLP syndrome?" I would think that blood pressure is the "crosstalk" between the 3 conditions. But the authors probably mean the crosstalk between the 3 clinical conditions and the two molecules. In any case, the formulation is unclear.
- Clinical hypertension (HTN) before pregnancy is usually called "chronic", "pre-existent" or "pregestational" hypertension. We are explained the definition of hypertension in pregnancy on page 8 (!) instead of the Introduction.
- Pre-eclampsia is defined on page 6. Again, we would expect to find this in the Introduction.
- The headings PAF and pre-eclampsia feature on page 4 and again on page 6, Mg and pre-eclampsia on page 5 and again on page 7.
Response: Insightful suggestions! Thank you for bringing this up, we have included in the introduction the definitions and parameters of clinical HTN and pre-eclampsia to give readers a better understanding prior to the later paragraphs. Additionally, we have reworked the titles of subsections in order to avoid the same subtitle being used twice. In regard to the “cross talk” aspect we have tried to make the connection clearer and more cohesive. In an attempt to improve the flow of our thought process we have rearranged the order in which we introduce the diseases as well as the molecules - and then bring together the actions of PAF and Mg. We hope this brings together the interplay between the two molecules as well as their individual as well as interdependent roles in the clinical conditions.
- Chronic HTN, pre-eclampsia and the HELLP syndrome are presented like different conditions, which obviously there are not. We would like to know when and how Mg and PAF come into the EVOLVING picture.
Response: Thank you for your review! We understand that connecting the two molecules in these clinical conditions is important, as per our response above - we have changed the order of discussion of the clinical conditions and the molecules in order to make the evolving picture clearer! Additionally, we have included a cross talk paragraph prior to the conclusion to further highlight the relationship.
- on Mg: "intravenous Mg therapy is used to counteract preeclampsia and its symptoms" (line 167) and "intravenous magnesium therapy is used to counteract HELLP and its symptoms" (line 185). Mg is used primarily to prevent or treat eclampsia, convulsions, and works at the motor neuron - muscle interface. Strangely for an article on Mg, the authors have made no inquiries on the rationale and the therapeutic history (introduced at Dallas, in fact) of Mg. "Counteract" should suffice.
Response: We appreciate this comment in regard to intravenous use of Mg. We have decided to include this in the discussion of pre-eclampsia as there is evidence that shows that intravenous Mg is utilized to keep symptoms at bay as well as further treat the hypertension seen. We have also elaborated on the therapeutic rationale for this use as well as efficacy of MgSO4. We have eliminated this sentence when it comes to HELLP syndrome as it requires more research to be conducted to determine the exact improvements that are found with intravenous Mg. Thank you for this insightful suggestion!
Round 2
Reviewer 3 Report
No further questions